# Uremic Toxins Induce THP-1 Monocyte Endothelial Adhesion and Migration through Specific miRNA Expression

**DOI:** 10.3390/ijms241612938

**Published:** 2023-08-18

**Authors:** Andres Carmona, Fatima Guerrero, Juan R. Muñoz-Castañeda, Maria Jose Jimenez, Mariano Rodriguez, Sagrario Soriano, Alejandro Martin-Malo

**Affiliations:** 1Maimonides Biomedical Research Institute of Cordoba (IMIBIC), 14004 Córdoba, Spain; andres.carmona@imibic.org (A.C.); mariajose.jimenez@imibic.org (M.J.J.); juanm.rodriguez.sspa@juntadeandalucia.es (M.R.); marias.soriano.sspa@juntadeandalucia.es (S.S.); amartinma@senefro.org (A.M.-M.); 2Department of Medicine, University of Cordoba, 14004 Córdoba, Spain; 3Nephrology Unit, Reina Sofia University Hospital, 14004 Córdoba, Spain; 4Redes de Investigación Cooperativa Orientadas a Resultados en Salud (RICORS), Instituto de Salud Carlos III, 28029 Madrid, Spain

**Keywords:** atherosclerosis, indoxyl sulphate, p-cresol, adhesion, migration, miRNAs

## Abstract

Atherosclerosis is initiated by the activation of endothelial cells that allows monocyte adhesion and transmigration through the vascular wall. The accumulation of uremic toxins such as indoxyl sulphate (IS) and p-cresol (PC) has been associated with atherosclerosis. Currently, miRNAs play a crucial role in the regulation of monocyte activation, adhesion, and trans-endothelial migration. The aim of the present study is to evaluate the effect of IS and PC on monocyte adhesion and migration processes in monocytes co-cultured with endothelial cells as well as to determine the underlying mechanisms. The incubation of HUVECs and THP-1 cells with both IS and PC toxins resulted in an increased migratory capacity of THP-1 cells. Furthermore, the exposure of THP-1 cells to both uremic toxins resulted in the upregulation of BMP-2 and miRNAs-126-3p, -146b-5p, and -223-3p, as well as the activation of nuclear factor kappa B (NF-κB) and a decrease in its inhibitor IĸB. Uremic toxins, such as IS and PC, enhance the migratory and adhesion capacity of THP-1 cells to the vascular endothelium. These toxins, particularly PC, contribute significantly to uremia-associated vascular disease by increasing in THP-1 cells the expression of BMP-2, NF-κB, and key miRNAs associated with the development of atherosclerotic vascular diseases.

## 1. Introduction

Uremic toxins, such as indoxyl sulphate (IS) and p-cresol (PC), are difficult to remove with conventional dialysis techniques because of their strong protein-binding capabilities [1]. Hence, these compounds accumulate in the blood of patients with advanced chronic kidney disease (CKD) [2]. In fact, uremic patients present a chronic microinflammatory state caused by the accumulation of toxins that promote endothelial dysfunction [3], leading to cardiovascular disease (CVD) [4] and other complications [5,6]. The effects of uremic toxins on monocyte adhesion and migration across the endothelium have been associated with the development and growth of the atherosclerotic plaque [7].

The vascular endothelium is the target of atherogenic factors, and endothelial dysfunction is considered an initial event in the development of atherosclerosis [8]. Monocytes are essential to the development and exacerbation of atherosclerosis [9]. In this context, monocytes are activated and acquire a pro-inflammatory and pro-atherogenic phenotype; the adherence to the endothelium is increased, which stimulates the development of atherosclerosis [10,11]. Therefore, being aware of the mechanisms underlying monocyte adhesion and transmigration on vascular endothelium cells is essential to prevent the development of atherosclerosis.

MicroRNAs (miRNA) are small non-coding RNA molecules that regulate gene expression post-transcriptionally [12]. Several studies have suggested that mature miRNA dysregulation plays an important role in the pathogenesis and progression of CKD [13,14]. Recent evidence strongly suggests that miRNAs play a critical role in the key pathophysiological mechanisms underlying the process of atherosclerosis [15].

To date, there is scarce evidence in the literature addressing the processes of monocyte adhesion and migration in cardiovascular disease associated with chronic kidney disease [16,17]. Furthermore, the mechanisms whereby uremic toxins exert their deleterious effects on this process are not fully known. The objectives of this study were the following: firstly, to evaluate the interaction between mononuclear cells and the endothelium in the presence of IS and PC; secondly, to identify molecular mechanisms induced by these two uremic toxins that may have an effect in the atherogenic process; and thirdly, to identify the potential differential effects of IS and PC.

## 2. Results

### 2.1. IS and PC Induce Cell Migration to the Endothelium

To investigate the migratory capacity of monocytes to the endothelium in a medium with these uremic toxins, the HUVECs monolayer and THP-1 pre-treated with IS or PC were co-cultured in a Transwell system. After 24 h, the migratory capacity of the THP-1 cells was assessed using fluorescence microscopy. We found that the treatment of cells with IS or PC induced a significant increase in the migratory capacity of THP-1 as assessed using fluorescence intensity (Figure 1A). Likewise, we also observed that the number of migrated THP-1 cells was increased if treated with either IS or PC (Figure 1B).

### 2.2. Effect of IS and PC on the Adhesion of THP-1 to Endothelium

Analysis of the effect of IS and PC on monocyte adhesion to the endothelium was achieved by performing a co-culture of HUVECs and THP-1 cells. After 30 min, THP-1 adhesion to the endothelial cells was assessed using fluorescence microscopy. Figure 2A shows the images of fluorescence microscopy, indicating that THP-1 cells exposed to uremic toxins have an increased capacity to adhere to the vascular endothelium. Then, fluorescence values (FU/Fields) were quantified, and it was found that the adhesion of PC-treated THP-1 cells was significantly increased (*p* < 0.01), compared to the other groups (Figure 2B).

### 2.3. The Expression of BMP-2 mRNA, miRNA-126-3p, miRNA-146b-5p, and miRNA-223-3p Are Increased in THP-1 Exposed to IS and PC

Bone morphogenetic protein 2 (BMP-2) is a cytokine that is elevated in the process of atherosclerosis and contributes to the vascular inflammation and calcification associated with uremia. In THP-1 cell culture, the addition of IS or PC causes a significant increase in BMP-2 mRNA expression compared to the control (*p* < 0.01 and *p* < 0.01, respectively). Interestingly, PC-treated THP-1 cells show a greater increase in BMP-2 mRNA compared to IS-treated cells (*p* < 0.01) (Figure 3A).

Moreover, recent data suggest that miRNAs are involved in the modulation of several key processes at all stages of the development of CKD-associated atherosclerosis. As shown in Figure 3B–D, the addition of IS or PC to THP-1 cells resulted in a marked increase in the expression levels of miRNAs-126-3p, -223-3p, and -146b-5p, compared to the control. Interestingly, it was observed that the expressions of miR-126-3p (*p* < 0.05) and -223-3p (*p* < 0.01) were significantly greater in THP-1 cells exposed to PC than in those exposed to IS. In addition, the non-significant numerical increase in miRNA-146b-5p expression levels in PC-treated vs. IS-treated cells (*p* = 0.07) should be mentioned.

### 2.4. IS and PC Regulate p65-NF-ĸB and IĸB Protein Expression in THP-1

The p65-NF-ĸB and its inhibitory protein, IĸB, play important role in the inflammation associated with CKD. We have assessed the expression of p65-NF-ĸB and IĸB in THP-1 cells treated with IS or PC using Western blotting (Figure 4A). As observed, the addition of IS or PC to THP-1 increased the p65-NF-ĸB levels compared to the control (*p* < 0.05). By contrast, we observed a significant reduction in IĸB expression in both groups compared to the control (*p* < 0.05). Again, a differential effect was observed between the two uremic toxins. Monocytes treated with PC presented a significantly greater increase in p65-NF-ĸB expression (*p* < 0.05) and less IĸB expression (*p* < 0.01) compared to IS (Figure 4B,C).

## 3. Discussion

Atherosclerosis is strongly linked to other inflammatory diseases, including CKD [18]. In fact, patients with CKD have an high risk of morbidity and mortality due to CVD, even during the early stages of the disease before uremic symptoms are developed [19]. This condition is characterized by systemic inflammation and alterations in leukocytes that are not well defined. In the present study, we have conducted experiments in an in vitro model to evaluate the adhesion and transmigration capacity of monocyte THP-1 cells exposed to two uremic toxins, IS and PC. Quantitative changes of the BMP-2 gene, miRNAs (miRNA-126-3p, miRNA-146b-5p, and miRNA-223-3p), and expression of p65-NF-ĸB and its inhibitor IĸB were observed. Our results have shown that IS and PC enhance the adhesion and transmigration ability of THP-1 cells and also induce the expression of BMP-2, p65-NF-ĸB, and specific miRNAs that are involved in monocyte adhesion and transmigration.

Atherosclerosis and CVD are closely related, in fact, atherosclerosis is a major cause of many cardiovascular diseases [20]. Atherosclerosis is notably present in patients with CKD, probably due to the accumulation of uremic toxins in the bloodstream [21]. Among these toxins, protein-bound uremic toxins, particularly IS and PC, that cannot be removed easily by conventional dialysis techniques have been associated with CKD progression and increased risk of CVD morbidity and mortality in uremic patients. Uremic toxins produce persistent inflammation, oxidative stress, and endothelial dysfunction resulting in accelerated atherosclerosis [22,23]. Recently, the diverse atherogenic effects of IS [24] have been highlighted, and other authors have reported that the serum levels of PC are positively correlated with the occurrence and severity of atherosclerosis [25].

One the earliest cellular events recognized so far in the development of the atherosclerotic lesion is the adherence of monocytes to the endothelium [26]. Several studies have reported that uremic toxins induce pro-inflammatory activation and adhesion of monocyte to the endothelium [22,27,28,29,30]. Monocytes cultured in vitro with uremic toxins generate various cytokines (IL-6, TNF-α, and IL-1β) [27,29,31] and adhesion molecules, such as VCAM-1 and ICAM-1, that are similar to those found in monocytes from CKD patients [32]. These effects clearly contribute to increased interactions between the leucocyte and the endothelium [25]. In the present study, we performed experiments using IS and PC at concentrations similar to those found in uremic patients, according to the data provided by the European Uremic Toxins Work Group (EUTox) [2], which report that the mean concentrations of IS and PC are 53.0 ± 91.5 mg/L and 20.1 ± 10.5 mg/L, respectively, with maximum levels of 236.0 mg/L for IS and 40.7 mg/L for PC. Considering that the concentration of IS and PC were comparable to blood levels in uremic patients, we have observed that both IS and PC promote the migration and adhesion of monocyte cells to activated vascular endothelial cells. In order to understand the molecular mechanisms by which uremic toxins modulate these processes, we have assessed the expression levels of miRNAs-126-3p, -146b-5p, and -223-3p for their potential roles in vascular disease and possible downstream targets, such as the transcription factor NF-ĸB and BMP-2.

The miRNAs regulate important aspects of vascular function, which are key in the development of atherosclerosis [33,34,35]. Some studies have reported a close association between the circulation of miR-223 and miR-126 and the occurrence of an atherosclerotic lesion [36,37]. In other chronic inflammatory diseases, such as axial spondyloarthritis, there has been an observed increase in expression levels of miR-223-3p and -126-3p in CD14+ monocytes [38]. In an in vitro study, an intracellular increase in miRNA-223-3p has been observed in THP-1 after being exposed to pro-inflammatory cytokines such as TNF-α [39]. The miRNAs regulate different molecular targets in the different inflammatory processes [40]. For instance, both miRNA-126-3p and -223-3p target IĸB, an inhibitor of NF-ĸB in its downstream pathway. Data indicate that overexpression of miR-126-3p decreases IκB expression in a dose-dependent manner [41]. Likewise, the decrease in miRNA-223 expression and the increase in IĸB expression resulted in a significant increase in NF-ĸB during monocytes differentiation into macrophages [42]. Accordingly, the results of the present study show that the upregulated expression of miRNAs-126-3p and -223-3p in THP-1 exposed to IS and PC induce NF-ĸB activity by inhibiting IĸB expression levels.

There is limited understanding of the molecular mechanisms whereby uremic toxins regulate the adhesion and migration of monocyte cells to vascular endothelium. Recently, Campillo et al. have demonstrated that, under IS and PC stimulation, monocyte cells increase podosome formation implicated in cell migration and invasion mediated by activation of integrin-linked kinase (ILK)/AKT signaling [17]. Also, activated ILK stimulates several signaling pathways including NF-ĸB [43]. Other studies have reported that the stimulation of monocytes with microparticles isolated from the plasma of patients with systemic lupus erythematosus favors the cell migration to peripheral tissues and perpetuate the inflammatory state mediated by the NF-κB pathway [44]. Furthermore, it is also known that NF-ĸB selectively modulates BMP-2 mRNA expression [45,46] and that BMP-2 exerts its pro-inflammatory effects by inducing monocyte migration and adhesiveness to endothelial cells, contributing to the development of atherosclerosis [47]. These studies support our findings, indicating that the upregulation of BMP-2—along with the elevation of NF-κB levels observed in THP-1 cells in a uremic environment—may promote the recruitment of monocytes.

Additionally, previous studies have reported that miR-146b expression is induced in both circulating monocytes [48] and the human monocyte cell line THP-1 by pro-inflammatory mediators such as LPS [49]. Our results have shown an increased expression of miR-146b-5p in THP-1 cells stimulated with IS and PC. We are aware of only one study related to cancer, showing that miRNA-146b-5p contributes to the regulation of cell migration and invasion in tumor cells [50]. Therefore, we do not know of any previous study showing the role of this miRNA on migration monocyte cells in CKD. Moreover, the expression of miRNA-146b-5p has been associated with cell senescence [51], and it is known that senescent monocytes increase their ability to interact with endothelial cells [10,52]. Consistent with these reports, an increased expression of miRNA-146b-5p observed in THP-1 stimulated with IS and PC could induce a senescence phenotype that favors the adhesion of monocyte cells to the vascular endothelium. This observation may, in part, elucidate possible mechanisms behind atherosclerosis development in CKD patients.

## 4. Materials and Methods

### 4.1. Preparation of Uremic Toxins

Indoxyl sulphate potassium salt (IS) and p-cresol 4-methylphenol (PC) (Sigma-Aldrich, St. Louis, MO, USA) were used. A stock solution of 0.397 M IS was prepared by dilution in distilled water. Working solutions were prepared by diluting the stock solution with culture medium in a 1:10 (*v*/*v*) ratio. PC was diluted in methanol to a final concentration of 0.0956 M. HUVECs and THP-1 cells were treated with IS (150 µg/mL) and PC (25 µg/mL) at concentrations within the reported uremic range as previously used in in vitro studies [53].

### 4.2. Cell Culture and Reagents

Human umbilical vein endothelial cells (HUVECs; Lonza, Walkersville, MD, USA) were cultured in standard endothelial basal medium (EBM; Lonza) supplemented with endothelial cell-growth medium additives (EGM, Lonza) and 10% fetal bovine serum (FBS; Invitrogen; Carlsbad, CA, USA). HUVECs from passages 4 to 9 were used for the experiments. HUVECs at 80% confluence were subjected to 24 h stimulation with or without IS or PC in 2% FBS conditions to maintain cellular quiescence. Human monocytes (THP-1; Sigma-Aldrich) were cultured in RPMI-1640 medium (Lonza) containing 1% penicillin–streptomycin, 2 mM glutamine, and 10% FBS. To maintain the cells in exponential growth, cultures had between 3–8 × 10^5^ cells/mL with culture media being refreshed every 2 days. For the experiments, THP-1 cells were treated with or without IS or PC during 48 h at 2% FBS. All the cells were cultured in a humidified incubator at 37 °C, 5% CO_2_, and 95% humidity.

### 4.3. THP-1 Adhesion Assay

A measurement of 2 µL of cell proliferation Dye eFluor 450 (10 mM, Thermo Fisher Scientific; Waltham, MA, USA) was added to 1 mL of the THP-1 cell suspension at 1 × 10^6^ cells/mL in an RPMI-1640 medium and incubated for 30 min in a humidified incubator. Thereafter, 3 × 10^5^ THP-1 cells were added to a monolayer of HUVECs seeded in a 6-well plate for 30 min in a humidified incubator, followed by two washes with a pre-warmed RPMI medium to remove unbound THP-1. Fluorescence from adhered THP-1 cells was measured using a Tecan Infinite^®^ 200 PRO Microplate reader (Tecan, Mannedorf, Switzerland) with excitation and emission wavelengths of 360/450 nm.

### 4.4. THP1 Migration Assay

To assess THP-1 cell migration, Dye eFluor 450 pre-labelled cells were seeded on top of 5 μm pore polycarbonate filter Transwell 24-well plates (HTS Transwell24 well permeable support; Corning, NY, USA) with 3 × 10^5^ HUVECs in the lower chamber. After 24 h of incubation at 37 °C in 5% CO_2_, the migratory capacity of THP-1 cells was evaluated using a Tecan Infinite^®^ 200 PRO Microplate reader with excitation and emission wavelengths of 360/450 nm. Cell counting was also used to analyze THP-1 cell migration in ten random fields with an inverted fluorescence microscope (Optika Microscopes, Ponteranica, Italy) using a 20× objective.

### 4.5. BMP-2, miRNA 126-3p, miRNA 146-5p, miRNA 191-5, and miRNA 223-3p Expression in THP-1

Total RNA was isolated from THP-1 cells and lysed using TRI-Reagent (Sigma-Aldrich). RNA pellets were washed with ethanol, and the dried pellets were resuspended in RNAse-free water. RNA quantification was performed using spectrophotometry (NanoDrop ND-1000 UV–Vis Spectrophotometer). The mRNA levels of bone morphogenetic protein 2 (BMP-2) were assessed with a quantitative real-time PCR (LightCycler^®^ 96; Roche Diagnostics, Basel, Switzerland). A SensiFAST SYBR ONE-STEP kit (Bioline) was used to quantify mRNA expression levels. The resulting mRNA expression values were normalized to the levels of GAPDH mRNA. BMP-2 (Forward 5′-AGG-AGG-CAA-AGA-AAA-GGA-ACG-GAC-3′ Reverse 5′-GGA-AGC-AGC-AAC-GCT-AGA-AGA-CAG-3′), GAPDH (Forward 5′-TGA-TGA-CAT-CAA-GAA-GGT-GGT-GAA-G-3′ Reverse 5′-TCC-TTG-GAG-GCC-ATG-TGG-GCC-AT-3′).

For miRNAs quantification in THP-1 cells, cDNA was synthesized from 100 ng of total RNA using a miRCury LNA RT-Kit (Qiagen, Hilden, Germany) according to the manufacturer’s protocol (PCR System 9700, GeneAmp, Applied Biosystems, Waltham, MA, USA). Each cDNA was diluted 30-fold, and 3 μL of this diluted cDNA was used as a template and was amplified using the miRCury LNA SYBR Green Kit (Qiagen). The assay ID used were: hsa-miR-126-3p (YP00204227, Qiagen), hsa-miR-146b-5p (YP00204553, Qiagen), hsa-miR-223-3p (YP00205986, Qiagen), and U6 snRNA (YP00203907, Qiagen). The amplification was performed in a LightCycler 96 Real-Time PCR System. The relative amount of each miRNA was calculated using the comparative threshold (Ct) method with ΔCt, Ct (miRNA)–Ct (U6 snRNA) in THP-1 cells.

### 4.6. Protein Extraction and Western Blot Analysis

Cellular extracts from THP-1 were prepared according to standard protocol [54]. Total protein concentration was determined using the colorimetric reagent of the Bradford assay (Bio-Rad Laboratories, Hercules, CA, USA). To determine specific protein contents, cytoplasmic and nuclear extracts (50 μg each) were separated in a 4–20% Ready Gel Tris-HCl gel (Bio-Rad Laboratories) and then transferred to a nitrocellulose membrane. The membranes were blocked in TTBS with 5% milk for 1 h at room temperature and incubated with primary antibodies, including anti-p65-NF-κB and IκB (both from Cell Signaling Technology, Danvers, MA, USA) and β-actin as a loading control (Santa Cruz Biotechnology, Dallas, TX, USA) overnight at 4 °C. Specific bands were visualized using ECL Western Blotting Detection System (Amersham Biosciences, Buckinghamshire, UK). Subsequently, the intensity of the bands was quantified through densitometry analysis using Image J software (version 1.52p) by measuring the integrated optical density (IOD).

### 4.7. Statistics

Values are expressed as mean ± SD of six independent experiments. The data normal distribution was tested with the Shapiro–Wilk test. Parametric comparisons were performed using the one-way ANOVA with post-hoc Bonferroni corrections for multiple comparisons. Mann–Whitney and Kruskal–Wallis tests were used for quantitative variables where appropriate. A value of *p* < 0.05 was considered to indicate statistical significance. Data were analyzed using SPSS Statistics software version 25.0 (SPSS, Inc., Chicago, IL, USA), and graphs were generated with GraphPad Prism 8.0 (GraphPad Software, La Jolla, CA, USA).

## 5. Conclusions

Our results indicate that both PC and IS enhance the adhesion and migration capacity of monocytes to the vascular endothelium. This effect is closely related to the alterations in the expression of specific molecules, including p65-NF-ĸB and BMP-2, which are regulated by distinct miRNAs. Remarkably, p-cresol exhibits a more pronounced effect compared with IS. These findings support the notion that the dysregulation of miRNAs may play a direct role in the development of vascular disease in CKD patients.

## Figures and Tables

**Figure 1 ijms-24-12938-f001:**
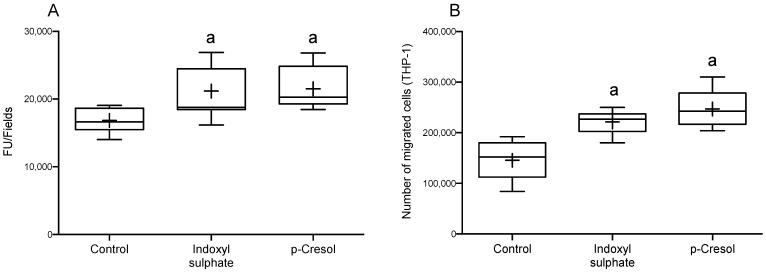
Indoxyl sulphate (IS) and p-cresol (PC) induced THP-1 cells migration to endothelial cells. THP-1 and HUVECs treated with or without IS (150 µg/mL) or PC (25 µg/mL). Both were co-cultured for 24 h. The migratory capacity of THP-1 cells was determined. (**A**) Assessing the migration capacity with fluorescence and (**B**) quantification of number of migrated cells. The data were analyzed using an ANOVA and post hoc Bonferroni test to evaluate statistical significance among groups. The bars extending from the boxes represent variability outside the upper and lower quartiles, while the mean is denoted by (+). The quantifications of migrated THP-1 cells are presented as the mean ± SD. ^a^ *p* < 0.01 vs. control.

**Figure 2 ijms-24-12938-f002:**
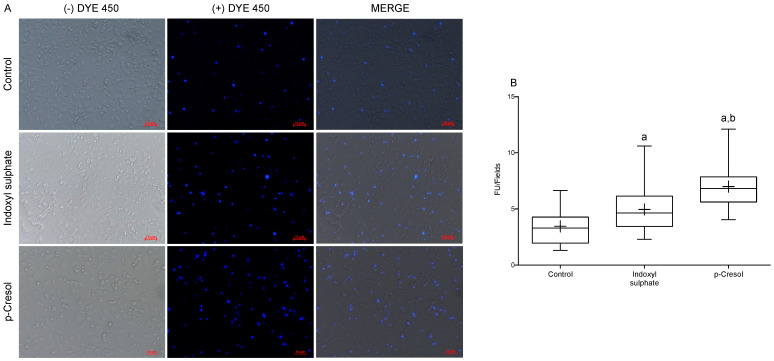
Indoxyl sulphate (IS) and p-cresol (PC) enhanced adhesion capacity of THP-1 to endothelial cells. THP-1 and HUVECs were treated with or without IS (150 µg/mL) or PC (25 µg/mL). Both were co-cultured for 30 min. (**A**) Representative images of fluorescence microscopy (20× magnification) of adhesion studies and (**B**) quantification of adherent THP-1 cells using fluorescence. The bars extending from the boxes indicate variability outside the upper and lower quartiles. The mean is represented as (+). The data were analyzed using an ANOVA and post hoc Bonferroni test to evaluate statistical significance between groups. The quantifications of adherent THP-1 cells are represented as the mean ± SD. ^a^ *p* < 0.01 vs. control; ^b^ *p* < 0.01 vs. indoxyl sulphate.

**Figure 3 ijms-24-12938-f003:**
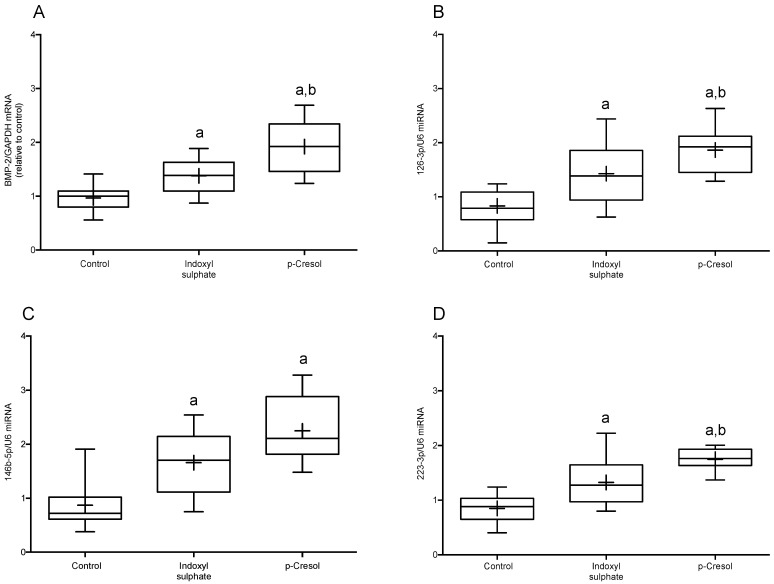
Expression levels of BMP-2 mRNA and miRNA-126-3p as well as miRNA-146b-5p and miRNA-223-3p in THP-1 cells. Box plot of (**A**) BMP-2, (**B**) miRNA-126-3p, (**C**) miRNA-146b-5p, and (**D**) miRNA-223-3p in THP-1. IS and PC induces differential expression of BMP-2, miRNA-126-3p, miRNA-146b-5p, and miRNA-223-3p in THP-1. The mean is represented as (+). The data were analyzed using an ANOVA and post hoc Bonferroni test to evaluate statistical significance among groups. Data are the means ± SD. ^a^ *p* < 0.05 vs. control; ^b^ *p* < 0.05 vs. indoxyl sulphate.

**Figure 4 ijms-24-12938-f004:**
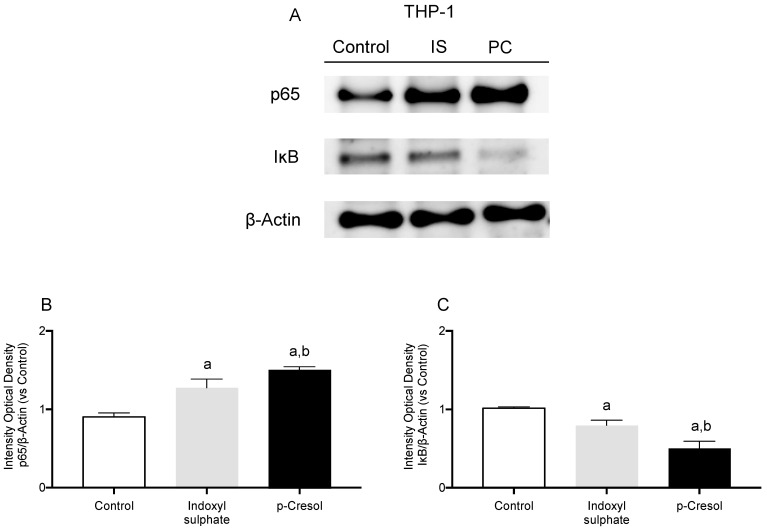
Indoxyl sulphate (IS) and *p*-cresol (PC) modify p65-NF-κB and IκB in THP-1 cells. (**A**) Representative Western blots of p65-NF-ĸB and IĸB. Densitometry analysis of blots for (**B**) p-65-NF-ĸB and (**C**) IĸB quantified using ImageJ. β-Actin was used as an internal control. IS and PC induced a significant increase in p-65-NF-κB expression and significant decrease in IκB expression in THP-1. The data were analyzed using an ANOVA and post hoc Bonferroni test to evaluate statistical significance between groups. Data are the means ± SD. ^a^ *p* < 0.05 vs. control; ^b^ *p* < 0.05 vs. indoxyl sulphate.

## Data Availability

The data used to support the findings of this study are available from the corresponding author upon request.

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
