# Peer review of "Uremic Toxins Induce THP-1 Monocyte Endothelial Adhesion and Migration through Specific miRNA Expression"

_ijms, 2023, doi:10.3390/ijms241612938_

Round 1

Reviewer 1 Report

The article is well written and contains interesting results about the uremic toxins and its correlation with miRNA.

The authors could add in the introduction or in the discussion, the concentration of uremic toxins used in vivo, in the circulation and compare that with the used dosage. The article states that the doses were calculated according to previous articles, but it can be repeated here, mainly discussing this in atherosclerosis. 

Reviewer 2 Report

- The authors aimed to evaluate the interaction between mononuclear cells and the endothelium; to identify molecular mechanisms induced by these two uremic toxins (IS and PC) in the atherogenic process; and finally, to investigate potential differential effects of IS and PC.

- The study Is novel and interesting and  fills a gap in the current literature.

- Methods are very well described.

- Results are relevant. Good figures and graphs.

- A discussion section is missing and should be added.

Author Response

We thank the reviewer for his/her comments. Discussion section is included in the new version of our manuscript (Page 5, line141).

Reviewer 3 Report

Since atherosclerotic progression is accelerated by chronic kidney disease (CKD), it has been postulated that uremic metabolites somehow potentiate the process.  The authors tested this hypothesis by conducting an in vitro study of the effects of two metabolites that accumulate in blood with impaired renal function, indoxyl sulfate and p-cresol, on human THP-1 monocytes.  They showed that the compounds promoted monocyte adhesion to and passage through human umbilical vein endothelial cell (HUVEC) monolayers and upregulate atherosclerotic and inflammatory markers and intermediates bone morphogenetic protein-1 (BMP-1), microRNAs miR126-3p, 146b-5p and 223-3p, and nF-kB, while downregulating the inhibitor of nF-kB, IkB.  This is a useful study that supports the hypothesis and lays the groundwork for an in vivo study to explore the effects of these metabolites on M1/M2 monocyte population balance, for instance.  The results also provide valuable corroboration of the studied microRNAs' pro-atherogenic effects.  It would have been preferable for the authors to have used a green fluorophore instead of eFluor 450, since the blue color is difficult to see in Figure 2.

Round 2

Reviewer 2 Report

amended manuscript is acceptable.